# Gender Composition in Occupational Therapy Journals’ Editorial Boards

**DOI:** 10.3390/ijerph20043458

**Published:** 2023-02-16

**Authors:** Cristina Mendoza-Holgado, Pablo A. Cantero-Garlito, Sabina Barrios-Fernandez

**Affiliations:** 1Health and Social Services Department, Government of Extremadura, 10004 Cáceres, Spain; 2Occupation, Participation, Sustainability and Quality of Life (Ability Research Group), Nursing and Occupational Therapy College, University of Extremadura, 10003 Cáceres, Spain; 3Faculty of Health Sciences, University of Castilla-La Mancha, 45600 Talavera de la Reina, Spain

**Keywords:** Occupational Therapy, journals, editorial boards, gender perspective

## Abstract

This paper analyses the Editorial Board (EB) distribution of Occupational Therapy journals from a gender perspective. The “Occupational Therapy” field in the Scimago Journal and Country Rank (SJR) and the “Occupational Therapy” term in the title search of the Journal Citation Report (JCR) were used to find the Occupational Therapy-specific journals. The following indicators were calculated: Editorial Board Member (EBM) gender distribution by journal, publisher, subject speciality, country, and journal quartile. Thirty-seven journals were located, including 667 individuals, 206 males (31%) and 461 females (69%). Referring to the EB positions, most members (557) were EB members, 70 were listed as Associate Editors, and 20 as Editorial Leaders. The results show that the proportion of women in the EB’s of Occupational Therapy journals represents a majority. Regarding the distribution by gender of the EBMs, six journals had a female proportion below the cut-off point revealed in this study (69%). Four did not reach parity, with female representation below 50%. Additionally, the balance among the EBMs is significantly underrepresented compared to the percentage of female Occupational Therapy practitioners.

## 1. Introduction

The Occupational Therapy profession has undergone remarkable changes in the last two decades worldwide, allowing more and more professionals to access academic and research spaces [1]. Thus, it is worth highlighting the substantial increase of women accessing relevant positions in these sectors in an eminently feminised profession. According to the World Federation of Occupational Therapists (WFOT) data provided by the 89 member organisations, 80.30% of Occupational Therapist practitioners are women. This percentage ranges from 100% in Jamaica, Seychelles, and Ukraine to 3% in the Dominican Republic. In healthcare professions, a glass ceiling limits women’s access to leadership and decision-making spaces [2]. Furthermore, there is a social perception that Occupational Therapy, like other health professions linked to care and mainly exercised by women, has a lower social value, and they are mainly focused on supporting Medicine [3]. In this sense, “the unequal distribution of power is also reflected in the institutions and structures related to science, and in turn, is reflected in scientific production” [4,5]. Health Science disciplines are constantly changing, so scientific works are becoming outdated faster. As Reale [6] points out: “a key concern of contemporary research policies is to demonstrate the ‘impact’ of research, or the value that public investment in research generates for increasing scientific competitiveness and excellence of the country, wealth creation, productivity, and social well-being”. Similarly, Brown [7] suggests that: “the Occupational Therapy literature is growing, and the citation frequency of articles is increasing”.

As Palser [8] notes: “journal editors wield considerable power over what is published and, by extension, over the direction of an academic discipline and the professional advancement of authors”. As an example, in the Spanish case, both in the academic and scientific fields, the aforementioned glass ceiling persists. Thus, according to data from the Spanish Government, women represent 42.20% of University Senior Lecturers, while on the Professor scale, the percentage decreases to 25.60%. Moreover, different qualifications between men and women also condition their access to other areas of science, such as the EB [9]. In this regard, the underrepresentation of women in the EB journals in Health Science disciplines is an important issue highlighted. Recent research [10] has shown that the gender gap in Medicine and Pharmacy journals is narrowing significantly; “although men still lag behind women in Nursing journals, they are and have been over-represented when considering the proportion of men practising in the field”. Recent work highlights the under-representation of women in EB journals in Health Science-specific journals, including 42% in Health journals [5], 35% in Medicine [10], 22% in Dermatology [11], 14.8% in Dental [12], and 33.05% in Paediatrics journals [13]. Although this study also claims that women’s participation was higher in Nursing, Physiotherapy, Occupational Therapy, Dermatology, and Nutrition journals. Thus, recent research suggests the need for efforts from multiple perspectives to address gender inequalities in Health Science journals [8,10].

The underrepresentation of women in EB of health scientific journals is a widely recognised issue and has multiple reasons. Firstly, there is a lack of equal opportunities and support for women in the scientific field, which can lead to a lower representation of women in leadership positions and contributions to high-impact articles [14]. Secondly, unconscious biases and gender stereotypes can play a role in the selection process for editorial board members, leading to a gender imbalance. Then, it is the result of multiple factors, including a lack of equal opportunities, gender biases, and stereotypes. Addressing these issues requires a multi-faceted approach, including efforts to promote gender equality, reduce bias, and increase the representation of women in leadership positions [15].

To our knowledge, this is the first study performed with a gender perspective on EBs in Occupational Therapy journals. As women constitute most of the Occupational Therapy practitioners worldwide, this study aims to identify and quantify the EBs’ distribution and gender differences in the Occupational Therapy journals included in the JCR (Journal Citation Reports) and the SJR (Scimago Journal and Country Rank).

## 2. Materials and Methods

### 2.1. Study Design and Search Strategy

This cross-sectional study included all the journals within the “Occupational Therapy” field in the SJR (https://www.scimagojr.com) (accessed on 4 June 2022); the search was completed using the JCR, including all journals with “Occupational therapy“ in the titles search on the Clarivate website (https://jcr.clarivate.com) (accessed on 4 June 2022). Because the information is publicly available herein, approval by an ethical committee is not required for this study.

### 2.2. Data Collection and Extraction

The search was performed in June 2022. The list formed by the journals included in the SJR and the JCR was merged, and duplicates were removed. The eligibility criteria were: (1) journals with a public EB, (2) Occupational Therapy discipline-specific, stated in their aims and scope, and (3) being in the Q1–Q4 quartile according to the SJR. A manual data inspection was conducted; any changes in the EBs made after this time were not included in the study. The journal’s web pages were audited. Because of the variability of the positions, they were categorised into three categories: (1) Editorial Leaders, (2) Associate Editors, and (3) EB Members, in line with other studies [8,10,16]. Moreover, the declared gender of all EB Members was determined by inspection of the authors’ names followed by a search on the internet. Finally, http://www.gender-api.com/ (accessed on 5 June 2022), a tool which estimates the percentage of certainty of a name’s gender, was used. A datasheet (Appendix A) was created with the following information: journal information extracted from the SCI (quartile, H-Index, country, region, publisher, coverage, categories) and the JCR, and information about every EB Member (position and gender).

### 2.3. Statistical Analysis

Data were compiled using Microsoft Excel (Microsoft Corporation, Redmond, DC, USA) to organise and manage the extracted information. Statistical analyses were carried out with the Statistical Package for the Social Sciences software version 25 (IBM SPSS, Armonk, NY, USA). Categorical variables were described with frequency and percentage. The EB male/female ratio was also obtained to improve the robustness of the results. The Chi-Square test with the Bonferroni correction was used to assess dependence relationships and differences in proportions among the groups (pairwise z-test for independent proportions) for categorical variables. Cramer’s V was used to check the strength of the association between the studied variables. Additionally, the Odds Ratio with a 95% Confident Interval was calculated for these variables. Statistical significance was considered when *p* < 0.05.

## 3. Results

### 3.1. Search Results

Firstly, Occupational Therapy journals worldwide were collected: 37 journals were located, 23 in the SJR and 14 in the JCR. From these, 12 were discarded because they were duplicates, and eight did not meet the eligibility criteria (Figure 1).

After the journals were identified, the EB’s information was sought. Different journals used the exact position to denote different seniority statuses and appear in various categories. Decisions were made on an individual basis, depending on the journal’s organisation. Table 1 displays the 21 positions found in the EBs’ composition, and then several categories were created to support the subsequent analyses using the categorisation model created by Palser et al. [8].

### 3.2. Gender-Related Results

A total of 667 individuals were found in the 17 journals: 206 males (31%) and 461 females (69%). The percentage of women obtained from the total number of journals was considered the cut-off point for the results of this study (Table 2).

Table 3 displays the Occupational Therapy journal’s information. According to the SJR, five journals were in the first quartile (Q1), six in the second (Q2), five in the third (Q3), and one in the fourth (Q4). There was a significant disparity between the number of EB members, being the lowest in the Canadian Journal of Occupational Therapy with nine and the highest in the Journal of Vocational Rehabilitation with 118 individuals. In terms of regional distribution, the journals were mainly distributed between Western Europe (35.29%) and the United States (47.06%), both considered western regions. The mean of members per journal was 39.24, and the median was 26. The mean of members per journal was 39.24, and the median was 26. Additional information on journals can be found in Appendix A.

Regarding the distribution by gender of the EB members, six journals had a female proportion below the cut-off point revealed in this study (69); four did not reach parity, with female representation below 50% (Figure 2).

Regarding the male/female ratio (Table 4 and Figure 3), the journals with the highest female participation were The Journal of Occupational Therapy, Schools, and Early Intervention (11.33) and Occupational Therapy in Mental Health (7.25); and those with the lowest, Physical and Occupational Therapy in Geriatrics (0.73) and The Hong Kong Journal of Occupational Therapy (0.72). More information about the male/female ratio can be found in Appendix A.

Referring to the positions within the EBs, 557 individuals were EB members, 70 were Associate Editors, and 20 were Editorial Leaders (Table 5).

According to gender, the smallest percentage of women were in Editorial Leadership positions, below the cut-off point (Figure 4).

Thus, considering the average number of women in journals as a descriptive analysis, it is worth noting that in terms of composition, the position of Editorial Leader was below the cut-off point, and this situation was repeated when considering the quartile in which Q1 journals have a percentage of women below the cut-off point. Moreover, the z-test found no significant differences between the proportions of the different editorial positions, quartiles, or regions (Table 6).

Furthermore, the Odds Ratio did not show a statistically significant value, agreeing with the contingency results shown in the binary logistic regression models by EB position obtaining a non-significant Odd Ratio for the gender variable in Editor Leader 0.686 (OR 0.825, 95%CI 0.324–2.009), Associate Editor 0.209 (OR 0.692, 95%CI 0.390–1.228), and Editor 0.353 (OR 1.267, 95%CI 0.769–2.087).

## 4. Discussion

### 4.1. Main Findings and Theoretical Implications

This study aimed to identify EB composition and quantify gender differences in the EBs of the Occupational Therapy journals in the SJR and JCR. According to our findings, females represent 69% of the EBs, a lower percentage than the data provided by the WFOT, which places the percentage of female Occupational Therapy practitioners at 80.30%. Only four of the seventeen journals identified exceeded the total percentage of Occupational Therapy practitioners, while six of them would maintain a female percentage even lower than the cut-off point obtained from the journals, of which four would not reach parity. The presence of women is higher in all established categories (Editorial Leader, Associate Editor, and EB Member), with the highest female representation in the Associate Editor position (74.28%). These results were consistent even when grouping the journals according to their quartile, with the highest women number in Q2 journals.

These results may be explained as a consequence of the Healthcare professions’ feminisation [17]. However, in previous studies in other Health Sciences disciplines, data were not in this line. For example, in Medicine, with significant research on this topic [18,19] and including specialities such as Anaesthesia [20,21], Surgery [22,23,24], Neurology [25], Paediatrics [13], Orthopaedics [26], Psychiatry [27], Dermatology [11], or Radiology [28,29,30] showed important gender inequalities with a widespread under-representation of women. Another study assessed the 21 most relevant journals in Medicine, Nursing, and Pharmacy, analysing their trends over 20 years (1995–2016) and finding that in Medical and Pharmacy journals, women were still behind their male counterparts on the EBs. In Nursing, men lag behind women, although there is an over-representation considering the proportion of male nurses [10]. One study revised the EBs of the top 50 journals in Psychology and Neuroscience, and significant differences were found regarding gender, with 76% of Psychology journals and 88% of Neuroscience journals having more than 50% male editors. In contrast, only 20% and 10% had a similar proportion of female editors [8]. Another manuscript analysing 143 Veterinary journals confirmed the gender disparity in this discipline since only 34.3% of the EBs investigated were female members [16]. Thus, 113 Rehabilitation and Sports Science journals were examined from a gender, geographic, and economic perspective. Their results showed a gender gap: between the 7248 editors found, only 24.7% were women; moreover, only 10.4% were Editorial Leaders. Furthermore, in this area, it was found that when a woman is the leader of an EB, there is a positive correlation with a higher presence of women in the EBs [31]. In line with the studies presented above, our results point out that women’s presence in the most relevant positions and quartiles is lower than in the rest of the categories and even below the cut-off point set by the average women’s presence in the EBs of the Occupational Therapy journals in this study. Previous research has identified that women are under-represented in EB leadership positions. The results in this study were in this line, showing fewer women in the “Editorial Leadership” positions (65% below the established cut-off point). However, this trend is reversed in the “Associate Editor” position. Thus, this might indicate female empowerment in the Occupational Therapy journals in contrast to the trends observed in other Health Sciences journals. Some studies have found differences between non-Western and Western regions in the variability of EB composition [32]. Nevertheless, in the case of the EBs of the Occupational Therapy journals, it can be highlighted that a greater presence of journals is from Western regions.

According to the WFOT [33], the female Occupational Therapy practitioners’ percentage was 89% in 2020, a significantly higher figure than in the EBs. Nevertheless, it is important to highlight that the EBs are circumscribed to the scientific field and do not cover Occupational Therapy practitioners in clinical or academic fields. Regarding the latter, there are still gender disparities, and discrepancies in women’s representation could be attributed to characteristics of the academic environment. No previous studies have addressed the gender distribution of Occupational Therapy academics; although there are studies which have analysed Occupational Therapy publications from a bibliometric perspective, to our best knowledge, the gender perspective has not been explored. One study about the scientific production of Doctoral Theses in Spain introduced the possible existence of vertical segregation since the presence of women decreases as the academic rank increases [34], a finding consistent with our results. Therefore, a masked gender gap may be present.

### 4.2. Practical Applications

Gender-sensitive research can also provide a more complete and accurate understanding of the issues being studied, leading to more effective solutions and policies. This research, with a gender perspective, helps to uncover and address the systemic barriers and biases that contribute to gender inequality and disadvantage. This study is the first to examine female representation in different positions on EB in Occupational Therapy journals. Our findings contribute to the growing body of knowledge showing how female Occupational Therapy practitioners are represented in discipline-specific journals. It is crucial to investigate the possible barriers faced by women Occupational Therapist scientists to promote, elevate, and support their scientific careers.

Furthermore, the practical applications of this research are diverse: (1) it provides knowledge on a topic that has not previously been studied in Occupational Therapy, (2) it offers a framework for future comparative or longitudinal studies. Moreover, (3) because Editors manage scientific journals, promoting research in their fields, journals should offer equitable opportunities for women to enhance the fields they cover to promote representative science and provide solutions for all citizens. In the case of Occupational Therapy, the profession is ethically constrained to adapt to the characteristics, needs, desires, and contexts of its patients, and within their interventions to encourage Occupational Justice, giving voice to people with different characteristics, and in this case, with different genders [35,36].

### 4.3. Limitations and Future Lines

This research has several limitations. As this is a cross-sectional study, it cannot be inferred as either an upward or a reversal trend. In this line, we attempted to obtain this trend, but this was impossible as many journals did not publicly have this information available on their websites. Although data from multiple sources were combined, and each name was verified by different sources, the gender assignment may be incorrect. In the case of the computer tool, the gender classification is linked to a statistical weighting based on the frequency of female and male name use, which may not always be accurate [37]. In this study, based on the data provided by the *genderapi*, there is a 5.91 margin of error with a minimum value of 54 and a maximum of 100. It is also possible that the gender assigned may be different from the gender with which the individual identifies, as this information would require questioning every subject. Although the most commonly recognised genders are male and female, many other gender identities exist [38]. The means used to infer gender are limited since a binary classification is assumed.

Regarding future lines, a longitudinal study should be conducted to better understand the EBs’ composition trends regarding gender. Thus, this study will serve as a framework for generating such knowledge in the future. Furthermore, the lack of knowledge of gender distribution in the Occupational Therapy academic field is a limitation for the interpretation of the results and a complete discussion of the real situation in this field. The use of the cut-off point facilitates the interpretation of the results of this study but constitutes a threat to the external validity of the study. Another future line would be to analyse the Occupational Therapists’ presence on the EBs of non-specific Occupational Therapy journals, examining them from a gender perspective. Moreover, research to know each member’s country of origin could be considered to identify whether there are differences between countries. Furthermore, the underlying causes of this gender inequality must be examined to establish strategies to promote women’s participation in the EBs of Health Sciences journals.

## 5. Conclusions

The results showed that the women’s proportion in Occupational Therapy Journals represents a majority. Nevertheless, the proportion among leadership roles in Occupational Therapy remains significantly underrepresented compared with the percentage of female Occupational Therapy practitioners.

It is still unclear whether these simply reflect the high proportion of females working as Occupational Therapy practitioners or whether gender-related bias exists. Further research should investigate the underlying causes for the lower presence of women in the EBs regarding the number of Occupational Therapist practitioners worldwide. Furthermore, measures should increase awareness of gender bias and encourage women’s promotion.

## Figures and Tables

**Figure 1 ijerph-20-03458-f001:**
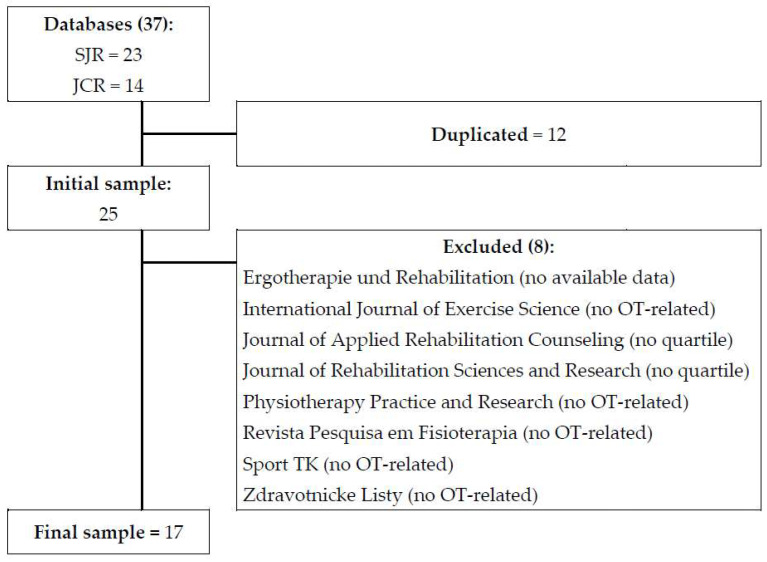
Flow diagram. SJR: Scimago Journal and Country Rank; JCR: Journal Citation Reports.

**Figure 2 ijerph-20-03458-f002:**
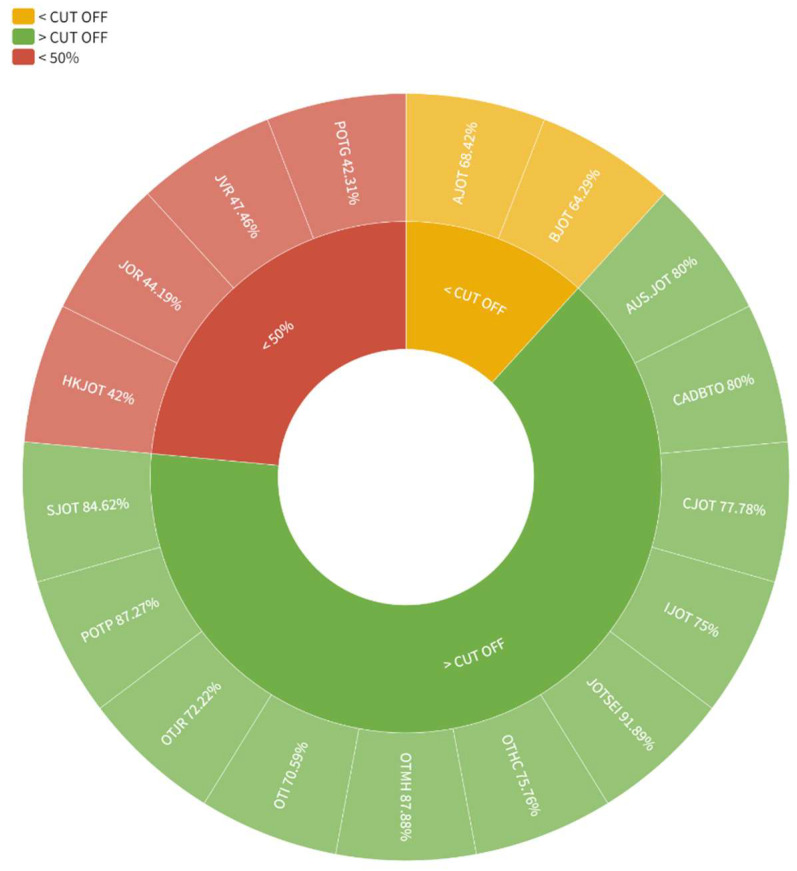
Distribution by gender of the editorial board members. JOTSEI: Journal of Occupational Therapy, Schools, and Early Intervention; OTMH: Occupational Therapy in Mental Health; POTG: Physical and Occupational Therapy in Geriatrics; SJOT: Scandinavian Journal of Occupational Therapy; AUSOTJ: Australian Journal of Occupational Therapy; CADBTO: Brazilian Journal of Occupational Therapy; CJOT: Canadian Journal of Occupational Therapy; OTHC: Occupational Therapy in Health Care; IJOT: Irish Journal of Occupational Therapy; OTJR: OTJR Occupation, Participation and Health; OTI: Occupational Therapy International; AJOT: American Journal of Occupational Therapy; BJOT: British Journal of Occupational Therapy; JVR: Journal of Vocational Rehabilitation; JOR: Journal of Occupational Rehabilitation; POTP: Physical and Occupational Therapy in Pediatrics; HKJOT: Hong Kong Journal of Occupational Therapy.

**Figure 3 ijerph-20-03458-f003:**
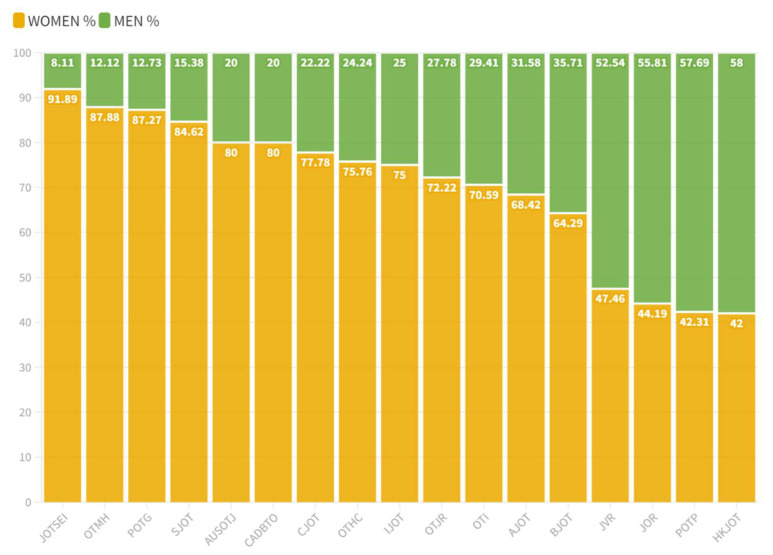
Gender composition of the editorial boards. JOTSEI: Journal of Occupational Therapy, Schools, and Early Intervention; OTMH: Occupational Therapy in Mental Health; POTG: Physical and Occupational Therapy in Geriatrics; SJOT: Scandinavian Journal of Occupational Therapy; AUSOTJ: Australian Journal of Occupational Therapy; CADBTO: Brazilian Journal of Occupational Therapy; CJOT: Canadian Journal of Occupational Therapy; OTHC: Occupational Therapy in Health Care; IJOT: Irish Journal of Occupational Therapy; OTJR: OTJR Occupation, Participation and Health; OTI: Occupational Therapy International; AJOT: American Journal of Occupational Therapy; BJOT: British Journal of Occupational Therapy; JVR: Journal of Vocational Rehabilitation; JOR: Journal of Occupational Rehabilitation; POTP: Physical and Occupational Therapy in Pediatrics; HKJOT: Hong Kong Journal of Occupational Therapy.

**Figure 4 ijerph-20-03458-f004:**
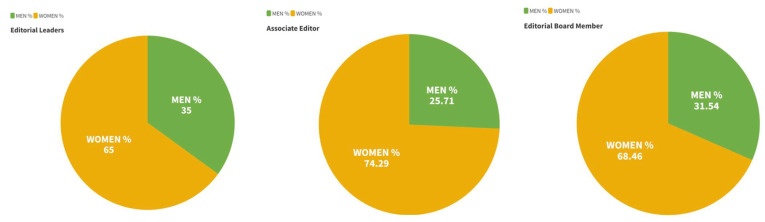
Composition of the editorial board according to gender.

**Table 1 ijerph-20-03458-t001:** Categories established for this study to group the different positions.

Categories	Journal Positions
Editorial Leaders	Editor in Chief, Editor, Co-Editors.
Associate Editor	Associate Editor, Senior Associate Editor, Editorial Associate.
Editorial Board Member	Editorial Board Member, Academic Editor, Managing Editor, Instructional Insights Assistant Editor, Founding Editor, Book Reviews, Technical Editor, Assistant Editor, National Editor, Evidence to Practice Editor, Section Editor, National Editorial Board, International Editorial Board, Assistant Editor in Chief, Editorial Advisory Board.

**Table 2 ijerph-20-03458-t002:** Gender description in the editorial boards of Occupational Therapy journals.

Gender	Number (%)
Men	207 (31%)
Women	460 (69%)
Total	667 (100%)

**Table 3 ijerph-20-03458-t003:** Characteristics of the Occupational Therapy journals.

Journal	SJR	JCR/JCI *	Q	H-Index	Country	Region	*n*	%
American Journal of Occupational Therapy	0.657	2.246 ^a^	Q1	87	United States	Northern America	19	2.85
Australian Occupational Therapy Journal	0.519	1.856 ^a^	Q1	47	United Kingdom	Western Europe	10	1.5
Brazilian Journal of Occupational Therapy	0.318	0.13 ^b^	Q3	7	Brazil	Latin America	55	8.25
British Journal of Occupational Therapy	0.372	1.243 ^a^	Q2	49	United Kingdom	Western Europe	14	2.1
Canadian Journal of Occupational Therapy	0.405	1.614 ^a^	Q2	58	United States	Northern America	9	1.35
Hong Kong Journal of Occupational Therapy	0.21	0.917 ^a^	Q3	15	Singapore	Asiatic Region	50	7.5
Irish Journal of Occupational Therapy	0.17	N/F	Q4	4	United Kingdom	Western Europe	16	2.4
Journal of Occupational Rehabilitation	1.049	3.489 ^a^	Q1	76	United States	Northern America	43	6.45
Journal of Occupational Therapy, Schools, and Early Intervention	0.262	0.22 ^b^	Q3	12	United Kingdom	Western Europe	37	5.55
Journal of Vocational Rehabilitation	0.409	0.41 ^b^	Q1	40	Netherlands	Western Europe	118	17.69
Occupational Therapy in Health Care	0.33	0.53 ^b^	Q2	26	United States	Northern America	33	4.95
Occupational Therapy in Mental Health	0.289	0.32 ^b^	Q3	20	United States	Northern America	66	9.90
Occupational Therapy International	0.354	1.448 ^a^	Q2	39	Egypt	Africa/Middle East	17	2.55
OTJR Occupation, Participation and Health	0.402	0.378 ^a^	Q2	42	United States	Northern America	18	2.70
Physical and Occupational Therapy in Geriatrics	0.254	0.26 ^b^	Q3	21	United States	Northern America	110	16.49
Physical and Occupational Therapy in Pediatrics	0.527	2.360 ^a^	Q1	48	United States	Northern America	26	3.9
Scandinavian Journal of Occupational Therapy	0.579	2.611 ^a^	Q2	44	United Kingdom	Western Europe	26	3.90
Total							667	100

* In those journals without JCR impact factor, the JCI was used; SJR: Scimago Journal Ranking; JCR ^a^: Journal Citations Report; JCI ^b^: Journal Citation Indicator; Q: quartile: %: percentage; JCI: Journal Citation Indicator; n: number; N/F: not found.

**Table 4 ijerph-20-03458-t004:** Bivariate analysis by gender of the editorial board members.

Journal	Men		Women		Male/Female Ratio
	*n* = 207	%	*n* = 460	%	
American Journal of Occupational Therapy	6	31.58	13	68.42	2.17
Australian Occupational Therapy Journal	2	20	8	80	4
Brazilian Journal of Occupational Therapy	11	20	44	80	4
British Journal of Occupational Therapy	5	35.71	9	64.29	1.80
Canadian Journal of Occupational Therapy	2	22.22	7	77.78	3.5
Hong Kong Journal of Occupational Therapy	29	58	21	42	0.72
Irish Journal of Occupational Therapy	4	25	12	75	3
Journal of Occupational Rehabilitation	24	55.81	19	44.19	0.79
Journal of Occupational Therapy, Schools, and Early Intervention	3	8.11	34	91.89	11.33
Journal of Vocational Rehabilitation	62	52.54	56	47.46	0.9
Occupational Therapy in Health Care	8	24.24	25	75.76	3.13
Occupational Therapy in Mental Health	8	12.12	58	87.88	7.25
Occupational Therapy International	5	29.41	12	70.59	2.4
OTJR Occupation, Participation and Health	5	27.78	13	72.22	2.6
Physical and Occupational Therapy in Pediatrics	14	12.73	96	87.27	6.86
Physical and Occupational Therapy in Geriatrics	15	57.69	11	42.31	0.73
Scandinavian Journal of Occupational Therapy	4	15.38	22	84.62	5.5

*n:* number; %: percentage.

**Table 5 ijerph-20-03458-t005:** Gender analysis of different editorial board member positions.

Position	Women		Men		Total	Male/Female Ratio
	*n* = 460	%	*n* = 207	%		
Editorial Leader	13	65	7	35	20	1.86
Associate Editor	52	74.28	18	25.71	70	2.89
Editorial Board Member	395	68.46	182	31.54	577	2.17

*n*: number; %: percentage.

**Table 6 ijerph-20-03458-t006:** Contingency table by genre and editorial position, quartile, journal, and region.

	Women		Men		X^2^	df	*p*	V
	n	%	n	%				
**Position**								
Editorial Leadership	13 ^a^	65	7 ^a^	35	0.1142	2	0.565	0.41
Associate Editor	52 ^a^	74.28	18 ^a^	25.71				
Editorial Board Member	395 ^a^	68.46	182 ^a^	31.54				
**Journal**								
American Journal of Occupational Therapy	13	68.42	6	31.58	109.040	16	<0.001	<0.001
Australian Occupational Therapy Journal	8 ^a^	80	2 ^a^	20				
Brazilian Journal of Occupational Therapy	44 ^a^	80	11 ^a^	20				
British Journal of Occupational Therapy	9 ^a^	64.29	5 ^a^	35.7				
Canadian Journal of Occupational Therapy	7 ^a^	77.78	2 ^a^	22.22				
Hong Kong Journal of Occupational Therapy	21 ^b^	42	29 ^a^	58				
Irish Journal of Occupational Therapy	12 ^a^	75	4 ^a^	25				
Journal of Occupational Rehabilitation	19 ^b^	44.19	24 ^a^	55.81				
Journal of Occupational Therapy, Schools and Early Intervention	34 ^b^	91.89	3 ^a^	8.11				
Journal of Vocational Rehabilitation	56	47.46	62	52.54				
Occupational Therapy in Health Care	25 ^a^	75.76	8 ^a^	24.24				
Occupational Therapy in Mental Health	58 ^b^	87.88	8 ^a^	12.12				
Occupational Therapy International	12 ^a^	70.59	5 ^a^	29.41				
OTJR Occupation, Participation and Health	13 ^a^	72.22	5 ^a^	27.78				
Physical and Occupational Therapy in Pediatrics	96 ^b^	87.27	14 ^a^	12.73				
Physical and Occupational Therapy in Geriatrics	11 ^b^	42.31	15 ^a^	57.69				
Scandinavian Journal of Occupational Therapy	22 ^a^	84.62	4 ^a^	15.38				
**Quartile**								
Quartile 1	192 ^b^	64	108 ^a^	36	6.738	3	0.098	0.081
Quartile 2	88 ^a^	75.21	29 ^a^	24.79				
Quartile 3	168 ^a^	71.79	66 ^a^	28.21				
Quartile 4	12 ^a^	75	4 ^a^	25				
**Region**								
Western	162 ^a^	29.7	383 ^a^	70.3	2.388	1	0.130	0.122
Non-Western	45	36.9 ^a^	77 ^a^	63.1				

*n*: number; %: percentage; X^2^: Pearson’s Chi-Square; df: degrees of freedom; *p*: *p*-value; V: Cramer’s V; a,b: different letters indicate significant differences between the proportions of editorial positions, journals, quartiles, regions, and sex at 95% z-test for independent proportions.

## Data Availability

Datasets will be available to the corresponding author under reasonable request.

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
