# Peer review of "Gender Composition in Occupational Therapy Journals’ Editorial Boards"

_ijerph, 2023, doi:10.3390/ijerph20043458_

Round 1

Reviewer 1 Report

This paper analyzed the editorial board distribution of occupational therapy journals from a gender perspective. The methodology is straightforward, and this paper is well-written. The authors did an excellent job reporting the research results. However, it would have been helpful if some gender issues related to journal editors had been addressed.

First, the authors found that the proportion of women in editorial leadership roles in occupational therapy remains significantly underrepresented. This point is worth mentioning, but one important factor that wans't mentioned is the gender distribution of academics in occupational therapy. I realize that such gender-related statistical information about academics is not expected to be found in databases. However, it is worth considering this issue.

Second, the key question that the authors could have investigated is, "Do female editorial boards have sufficient academic qualifications?" This is a highly relevant question, since the results may explain why there are fewer female editorial leaders than male editorial leaders. Editorial leaders are expected to have a high level of academic qualifications. Thus, the investigation of academic ranks, occupations, and the professional position of the editorial board is essential in this type of research. 

Third, it would be helpful to consider the country of the publishers when examining editorial leaders and editorial boards because some countries have a reputation for struggling with gender equality. In this regard, the authors may want to consider the following paper:

Oh, D. G., Kim, E., Yeo, J., Yang, K., & Lee, J. (2019). A Comparative Analysis of Editorial Leaders' Profiles of Major and Non-Western Library and Information Science Journals. Journal of Information Science Theory and Practice, 7(4), 20-32. https://jistap.accesson.kr/v.7/4/20/7411

Other minor points are the following:

In the title, the word "composition" should be capitalized for consistency.

What does 'no topic' mean in Figure 1? Does it mean the journal does not deal with occupational therapy topics? Justifiable statements are needed since some in the "no topic" category were removed from the research data.

In sum, authors should consider various points I mentioned to improve this paper.

Author Response

This paper analyzed the editorial board distribution of occupational therapy journals from a gender perspective. The methodology is straightforward, and this paper is well-written. The authors did an excellent job reporting the research results. However, it would have been helpful if some gender issues related to journal editors had been addressed.

Dear Reviewer,

Thank you very much for your review, comments, and suggestions, which have greatly contributed to improve the quality of the manuscript. Below, you can find our answers to your comments.

First, the authors found that the proportion of women in editorial leadership roles in occupational therapy remains significantly underrepresented. This point is worth mentioning, but one important factor that wans't mentioned is the gender distribution of academics in occupational therapy. I realize that such gender-related statistical information about academics is not expected to be found in databases. However, it is worth considering this issue.

Authors’ response: Thank you very much for your comment which has allowed us to reflect on this issue. We have included this consideration both in the introduction and in the discussion sections.

Second, the key question that the authors could have investigated is, "Do female editorial boards have sufficient academic qualifications?" This is a highly relevant question since the results may explain why there are fewer female editorial leaders than male editorial leaders. Editorial leaders are expected to have a high level of academic qualifications. Thus, the investigation of academic ranks, occupations, and the professional position of the editorial board is essential in this type of research. 

Authors’ response: You raise a very relevant issue. However, not all journals offer this information about their EBs, which would be fundamental to construct a well-founded discourse on qualifications. Given the lack of previous studies in this area, we believe that this first approximation may help us to find more defined lines of research in the future. Nevertheless, to underline the issue of qualifications, we have included in the introduction information that contextualises, by way of example, the academic disparity in our immediate setting (lines 69-75). 

Third, it would be helpful to consider the country of the publishers when examining editorial leaders and editorial boards because some countries have a reputation for struggling with gender equality. In this regard, the authors may want to consider the following paper:

Oh, D. G., Kim, E., Yeo, J., Yang, K., & Lee, J. (2019). A Comparative Analysis of Editorial Leaders' Profiles of Major and Non-Western Library and Information Science Journals. Journal of Information Science Theory and Practice, 7(4), 20-32. https://jistap.accesson.kr/v.7/4/20/7411

Authors’ response: Thank you. Both countries and regions have been incorporated in Table 3. A paragraph has been added on lines 181-184 and these findings have been considered in Table 6. The following paragraph has been added to the discussion “Some studies have found differences between non-Western and Western regions in the variability of EB composition [37]. Nevertheless, in the case of the EBs of the Occupational Therapy journals it can be highlighted the greater presence of journals from Western regions.”

Other minor points are the following:

In the title, the word "composition" should be capitalized for consistency.

Authors’ response: Amended.

What does 'no topic' mean in Figure 1? Does it mean the journal does not deal with occupational therapy topics? Justifiable statements are needed since some in the "no topic" category were removed from the research data.

Authors’ response: Thank you; “no topic” has been replaced by “no OT-related”.

In sum, authors should consider various points I mentioned to improve this paper.

Authors’ response: Thank you for taking the time to help us improve the quality of our manuscript.

Reviewer 2 Report

File attached.

Author Response

Reviewer 2

Thank you to the journal for having me as a reviewer. Dear authors, I am aware of the effort it takes to conduct research. My intention is to help you to improve your manuscript.

Dear Reviewer,

Thank you very much for your review, comments, and suggestions, which have greatly contributed to improve the quality of the manuscript. Below, you can find our answers to your comments.

Summary of the review

The study is relevant but needs to improve the introduction, properly substantiate the choice of its cut-off point (or even change the statistical approach) and provide explanations of the results in the discussion that generate new knowledge.

Authors’ response: Thank you. We have tried to consider all your suggestions.

Introduction

The introduction highlights the low representation of women in editorial teams even in female-dominated journals, however, there is no mention of the evolution of this event or possible explanations for it. It would be advisable to include a paragraph on the subject to make the reader focus more on the problem (theoretical framework), as well as to lengthen the introduction a little more.

Authors’ response: The introduction has been modified and expanded. The information on impact metrics, the peculiarities of the field of occupational therapy, as well as some of the reasons and possible solutions to the problem posed have been expanded. In addition, the explanation of the objective of the study has been expanded.

Materials and Methods

You take as a cut-off point to compare the results the percentage of the total number of women obtained in the same study, therefore, the comparison with the gender distributions of the selected journals is only comparable with your results (external validity is compromised). The percentage comparison would have to be provided according to the gender composition of occupational therapists according to the country of the journal in question or in general: if you say "According to the WFOT , the female Occupational Therapy practitioners' percentage was 89% in 2020", it is logical to think that EBs reach at least the same percentage, or be based on another way that would have to be defended by the researchers. The fact that EBs are more related to the academic field rather than the clinical field is already a gender gap in itself, and does not explain the reason for the differences found.

Authors’ response: We understand your concern about the threat to the external validity of the study and incorporate this in the limitations section. In the absence of previous studies, this research aims to have an approach to the topic of study that helps to map Occupational Therapy journals. Regarding the use of the cut-off point obtained from the gender composition of the journals in our sample, we believe that it facilitates the analysis within the context we are considering, given that we understand that the distribution of the composition of Occupational Therapy journals does not have to be circumscribed to that of occupational therapists. In the case of having used the population of each country as a comparison, we would be faced with the same problem with the addition that EBs tend to incorporate specialists from different geographical areas. For this reason, it seems methodologically appropriate to use the cut-off point of the real sample we are using, without losing sight of the total population data we are handling and assuming the aforementioned limitation.

The tables should show the sample size values per group. Low sample sizes are seen in some categories.

Authors’ response: Thank you. Tables have been modified including the n.

Discussion

You suggest that no gender differences were found since women represent 69.1% of the EBs, but then you say that "our results indicate that the presence of women in the most relevant positions and quartiles is lower than in the rest of the categories and even below the cut-off point marked by the average presence of women in the EBs of the Occupational Therapy journals in this study". This is confusing.

Authors’ response: Thank you. It was a mistake. We have amended it with: “According to our findings, females represent 69% of the EBs a lower percentage than the data provided by the WFOT, which places the percentage of female Occupational Therapy practitioners' at 80.30%. Only 4 of the seventeen journals identified exceed the total percentage of Occupational Therapy practitioners, while 6 of them would maintain a female percentage even lower than the cut-off point obtained from the journals, of which 4 would not reach parity.”

In the limitations you highlight that this is the first study conducted with a gender perspective on EBs in Occupational Therapy Journals. This is a strength rather than a limitation and it is usual to add "to the best of our knowledge".

Authors’ response: You are right. We have moved these lines to the introduction, where the study's aim is described. Thank you.

The genderapi application provides percentages that should have been used to weight the % error in assigning sexes according to names.

Authors’ response: Thank you. This information has been added in lines 343-344.

The possible self-selection bias may be in people describing their gender when they identify with another gender. This bias could have been overcome by including "declared" gender in the selection criteria.

Authors’ response: Thank you. Amended.

Round 2

Reviewer 1 Report

Most of the comments that were raised by this reviewer have been addressed by the authors in this revision.

Author Response

Dear Reviewer, thank you very much for helping us to improve the quality of our paper.

Reviewer 2 Report

Thank you for your clarifications. The following points should be reviewed.

There is no need to explain what the impact factor is or the various metrics that exist, in isolation, which have been included in lines 47 to 64.

The paragraph on lines 84-93 is repetitive:: “The underrepresentation of women in EB of health scientific journals is a widely recognized issue and has multiple reasons” and “The underrepresentation of women 89 in EB of scientific journals is a result of multiple factors”. This paragraph should be consolidated.

The paragraph from lines 98 to 101, after the statement of the objective, is more of a conclusion or implications for practice.

Table 2 is misaligned.

The statistical approach remains flawed. A contingency analysis with so many different journal categories would have benefited from a Z-test of two proportions, for example, which would have given an insight into the sex difference for each journal (also tables 4 and 6 could have been unified).

A parenthesis is missing in line 267.

In table 6, Region, the values of the columns by gender do not reach the required n-value. Revise the values.

---

Author Response

Thank you for your clarifications. The following points should be reviewed.

Dear Reviewer, below, you can find our answers to your comments.

There is no need to explain what the impact factor is or the various metrics that exist, in isolation, which have been included in lines 47 to 64.

Authors’ response: We have deleted that part. Thank you.

The paragraph on lines 84-93 is repetitive: “The underrepresentation of women in EB of health scientific journals is a widely recognized issue and has multiple reasons” and “The underrepresentation of women 89 in EB of scientific journals is a result of multiple factors”. This paragraph should be consolidated.

Authors’ response: Amended. Thank you.

The paragraph from lines 98 to 101, after the statement of the objective, is more of a conclusion or implications for practice.

Authors’ response: It has been moved to the practical applications in the discussion section. Thank you.

Table 2 is misaligned.

Authors’ response: Amended.

The statistical approach remains flawed. A contingency analysis with so many different journal categories would have benefited from a Z-test of two proportions, for example, which would have given an insight into the sex difference for each journal (also tables 4 and 6 could have been unified).

Authors’ response: Thank you very much for your comment. We have added the information you suggested, expanding the methodology sections and table 6 in the results. Since table 6 is now larger, we believe it is better not to merge tables 4 and 6 to make the content easier to read for readers.

A parenthesis is missing in line 267.

Authors’ response: Amended.

In table 6, Region, the values of the columns by gender do not reach the required n-value. Revise the values.

Authors’ response: Amended. Thank you.